# Development of Biocompatible Polyhydroxyalkanoate/Chitosan-Tungsten Disulphide Nanocomposite for Antibacterial and Biological Applications

**DOI:** 10.3390/polym14112224

**Published:** 2022-05-30

**Authors:** Abdul Mukheem, Syed Shahabuddin, Noor Akbar, Irfan Ahmad, Kumar Sudesh, Nanthini Sridewi

**Affiliations:** 1Department of Maritime Science and Technology, Faculty of Defence Science and Technology, National Defence University of Malaysia, Kuala Lumpur 57000, Malaysia; mukheembio@gmail.com; 2Department of Chemistry, School of Technology, Pandit Deendayal Energy University, Raisan, Gandhinagar 382426, Gujarat, India; 3Department of Biology, Chemistry, and Environmental Sciences, American University of Sharjah, Sharjah P.O. Box 26666, United Arab Emirates; noormicrobiologist555@gmail.com; 4Department of Clinical Laboratory Sciences, College of Applied Medical Sciences, King Khalid University, Abha 62529, Saudi Arabia; irfancsmmu@gmail.com; 5Applied Microbiology and Ecobiomaterial Research Laboratory, School of Biological Sciences, Universiti Sains Malaysia, Penang 11800, Malaysia; ksudesh@usm.my

**Keywords:** tungsten disulfide, polyhydroxyalkanoate, chitosan, biocompatibility, antibacterial

## Abstract

The unique structures and multifunctionalities of two-dimensional (2D) nanomaterials, such as graphene, have aroused increasing interest in the construction of novel scaffolds for biomedical applications due to their biocompatible and antimicrobial abilities. These two-dimensional materials possess certain common features, such as high surface areas, low cytotoxicities, and higher antimicrobial activities. Designing suitable nanocomposites could reasonably improve therapeutics and reduce their adverse effects, both medically and environmentally. In this study, we synthesized a biocompatible nanocomposite polyhydroxyalkanoate, chitosan, and tungsten disulfide (PHA/Ch-WS_2_). The nanocomposite PHA/Ch-WS_2_ was characterized by FESEM, elemental mapping, FTIR, and TGA. The objective of this work was to investigate the antimicrobial activity of PHA/Ch-WS_2_ nanocomposites through the time–kill method against the multi-drug-resistant model organisms *Escherichia coli* (*E. coli*) K1 and methicillin-resistant *Staphylococcus aureus* (MRSA). Further, we aimed to evaluate the cytotoxicity of the PHA/Ch-WS_2_ nanocomposite using HaCaT cell lines by using a lactate dehydrogenase (LDH) assay. The results demonstrated very significant bactericidal effects of the PHA/Ch-WS_2_ nanocomposite, and thus, we hypothesize that the nanocomposite would feasibly suit biomedical and sanitizing applications without causing any adverse hazard to the environment.

## 1. Introduction

Significant development in the field of nanotechnology and its biomedical application has been achieved through the synthesis of various biocompatible nanomaterials. The excellent physiochemical properties of nanomaterials such as graphene have sparked interest in two-dimensional (2D) nanomaterials [1]. Graphene is one of the most influential 2D nanomaterials, used in a wide variety of applications, including antibacterials, biomedical devices, and tissue engineering. The shapes of 2D nanomaterials possess high surface-area-to-volume ratios and other properties, such as biocompatibility, superior mechanical ability, flexibility, lubricity, photoluminescence, and catalytic activity, as compared to 0D, 1D, and 3D forms of carbon [2,3,4,5]. The use of graphene in biomedical applications, such as antibacterials, anticancers, biosensors, and nano-drug-carriers [1,2], recently prompted an additional search for 2D nanomaterials. Among various 2D nanomaterials, WS_2_ is broadly explored because of its outstanding nontoxic properties, chemical inertness, and structural stability, with a high melting point. These characteristics of WS_2_ have been explored in various applications, such as high-energy batteries, field-effect transistors, integrated circuits, phototransistors, sensors, drug carriers, gene-delivery vehicles, and anticancer platforms [6,7,8,9,10]. The tribological properties of WS_2_ have supported the coating of many medical devices, such as catheters, stents, and endodontic files, including artificial joints for lowering friction [11,12,13,14]. Redlich et al., reported that the incorporation of WS_2_ nanoparticles with stainless steel wires reduced 54% of the frictional force in orthodontic applications [15]. The biocompatibility of nanoparticles was examined based on three important aspects: their effect on cell proliferation, morphology, and direct interaction with the cells [11]. Jennie et al., demonstrated that WS_2_ showed limited cytotoxicity and genotoxicity, with no significant effect on cell viability [16]. In particular, the antibacterial abilities shown by 2D nanomaterials have drawn the attention of material scientists and biologists. The bactericidal activity of WS_2_ against the multi-drug resistant (MDR) model organism *E. coli* and MRSA were examined by the morphological changes in bacterial cell structure and by the quantification of reactive oxygen species (ROS) formation [17]. The bactericidal activity is feasibly due to the direct contact of the cell membrane with the sharp edges of WS_2_. This disrupts the cell membrane and causes oxidation stress, which imposes bacterial death, as shown in the schematic illustration, Figure 1. The effective antibacterial and biocompatible properties of WS_2_ allowed for the design of novel nanocomposites with suitable biopolymers for biomedical, therapeutic, and sanitizing applications. The combination of biopolymers and nanomaterials has led to a quantum leap in the advancement of drug-delivery systems and nanocomposite scaffolding.

Biopolymers play a significant role in sustainable drug-release systems and the fabrication of suitable drug-delivery scaffolds. Polyhydroxalkanoate (PHA) is a biopolymer synthesized in unbalanced growth conditions by more than 300 species of Gram-positive and Gram-negative bacteria as their intracellular carbon and energy storage compounds [18]. A few commonly used synthetic polymers fail to meet biodegradation and medicinal requirements, for example, silicones that cause cancer [19]. In comparison, PHA is considered an excellent biodegradable material because the degradation products of PHA are common metabolites in macro-organisms [20]. However, its brittleness and time-consuming synthesis make PHA an inadequate replacement for toxic synthetic polymers. Recently, many co-monomers of PHA have been biosynthesized to overcome these imperfections and attain desirable properties for application in the fields of polymer chemistry and biology. Its biocompatibility, biodegradability, and nontoxic nature have made PHA a flexible candidate, with a number of household, industrial, biomedical, and tissue-engineering uses [18]. Members of the PHA families P(4HB), P(3HB-3HHx), P(3HB), P(3HO), and P(3HB-3HV) are frequently used in a wide range of medical applications, such as nanocomposites, drug delivery, and implants [21]. PHA biomaterial has a critical role as a structural component of the extracellular matrix, with essential functions in tissue morphogenesis, cell attachment, and proliferation [22]. To further its usefulness, the adaptive properties of the PHA structure can be changed by chemical modification and physical blending. The intermingling of different biopolymers and 2D nanoparticles may feasibly result in a potential nanocomposite with complex functionalities for biomedical applications.

Chitosan is a biopolymer that has gained increasing attention in the scientific world because of its hydrophilicity, biocompatibility, and biodegradability. Chitosan is a natural polycationic linear polysaccharide derived from chitin [23,24,25,26]. Chitosan is a biodegradable and inexpensive polymer that has numerous applications in the biomedical and pharmaceutical industries [27]. In tissue engineering, chitosan and its derivatives are widely investigated, because they will be degraded in vivo by several proteases without producing any toxic end-products when new tissues are formed. Furthermore, its mechanical and structural properties support the proper functioning of repaired tissues. Alves and Mano et al., demonstrated that the cationic property of chitosan is responsible for the electrostatic communication with anionic glycosaminoglycans (GAGs), proteoglycans, and other negatively charged molecules that play a key role in the formation of tissue [28]. Moreover, chitosan, as a drug carrier, possibly improves drug absorption, stabilizes drug constituents, and controls drug-release augmentation [29]. However, the inferior mechanical properties of chitosan have limited its widespread application. Based on the abilities mentioned above, PHA and chitosan have mutual complementary potentials. Therefore, it is sensible to hypothesize that their separate deficiencies could be overcome by combining PHA and chitosan polymers to improve the potential for biomedical application [30].

In this paper, we investigated a nontoxic, cost-effective, fast, and eco-friendly approach for the synthesis of PHA/Ch, PHA/Ch-WS_2_ nanocomposites. The PHA and Ch blends were prepared using a single solvent glacial acetic acid reflux method. The average size of WS_2_ nanoparticles used in the nanocomposite was around 80 nm. Furthermore, the investigation was focused on the antibacterial and cytotoxicity analyses of the resultant nanocomposite. The antibacterial study was performed by time–kill method using five different time intervals to observe the time-dependent bactericidal activity of fabricated nanocomposites with three different wt% WS_2_ contents against MDR *E. coli* K1 and MRSA. LDH assays were employed to evaluate the cytocompatibility efficiency of PHA/Ch-WS_2_ nanocomposites against spontaneously immortalized human keratinocytes (HaCaT) cell lines. Thus, this research presents a simple, facile, cost-effective, and greener route for the fabrication of biopolymer-based nanocomposites, which can be effectively utilized in many biomedical applications, such as wound-healing scaffolds, antibacterial scaffolds, and sanitization materials.

## 2. Materials and Methods

### 2.1. Materials

WS_2_ powder, having an average nanoparticle size of 80 to 100 nm, was purchased from Lower Friction Company (Mississauga, ON, Canada). P(3HB-*co*-12 mol% HHx) powder (350,000 Da) was obtained from KANEKA Corporation, Osaka, Japan. Medium molecular weight chitosan powder was purchased from Sigma-Aldrich, St. Louis, MO, USA. Glacial acetic acid (99.8%) was obtained from Sigma-Aldrich, Petaling Jaya, Malaysia. Bacterial cultures of *E. coli* K1 and MRSA were grown and incubated for up to 16 h prior to antibacterial analysis. LDH kit was purchased from Roche Diagnostics, Indianapolis, IN, USA. Non-essential amino acid was purchased from Life Technologies, Carlsbad, CA, USA. In this experiment, analytical-grade reagents were used.

### 2.2. Preparation of Precursor Solution

An amount of 10 g of bare PHA was added to 150 mL of a round-bottom flask containing 10 mL of glacial acetic acid to obtain a final concentration of 1 g/mL. A magnetic stirrer plate was used to mix PHA thoroughly in glacial acetic acid prior to dissolution. Furthermore, silicon oil was heated to 118 °C, and the temperature was monitored using an external thermal probe. A typical reflux setup containing a simple cooling system was used to control the evaporation and retain the glacial acetic acid in the reaction without loss. PHA was dissolved by dipping the solution into the preheated silicon oil and allowed to stir for 5 min at 300 rotations per minute (rpm). WS_2_ nanoparticles at three concentrations (0.1 mg, 0.5 mg, and 1 mg) were added separately to their respective precursor solutions and mixed using a magnetic stirrer. To disperse the nanoparticles thoroughly, the PHA solution was bath sonicated for 30 min. To improve the wettability, chitosan (chitosan solution with a concentration of 1 mg/mL was prepared in 1% acetic acid) was added to obtain a 10:1 ratio of PHA/Ch.

### 2.3. Casting Film

The casting stage was prepared by using a medium-size flat glass plate with respect to the size of the hot plate. The thickness and size of the nanocomposites were controlled through standard microbiological glass slides of a 75 mm × 25 mm size. The casting stage was heated up to 80 °C, the precursor solution was then carefully dispensed onto it, and the temperature was maintained to evaporate the solvent. The solvent was completely evaporated between 7 and 10 min, leaving the films. The fabricated nanocomposites were peeled out by removing the microbiological glass slides from any one corner. The resultant nanocomposites were dried at room temperature to ensure complete solvent elimination prior to characterization and antibacterial analysis.

### 2.4. Antibacterial Assay

Time–kill technique was used to evaluate the bactericidal activity of the PHA/Ch-WS_2_ nanocomposites against the multidrug-resistant organisms *E. coli* K1 and MRSA. Generally, time–kill method is concentration- or time-dependent and was the most appropriate technique to determine the bactericidal effect [31]. This study used both the time-dependent method, at 0, 2, 4, 6, and 24 h, and the concentration-dependent method, at WS_2_ 0.1 mg, 0.5 mg, and 1 mg, to evaluate the PHA. All the experiments were repeated with two duplicates, and the obtained results were averaged statistically. A loop-full of pure culture broth was inoculated into the freshly prepared nutrient broth and incubated at 37 °C temperature overnight. To obtain c.1 × 10^8^ CFU/mL, the pre-activated bacterial culture was measured to 0.22 optical density at 595 nm. The nanocomposites with 0.6 mm disc thickness were dipped into the 100 µL nutrient broth, and then the 10 µL bacterial cultures were added to the density of c.1 × 10^6^ CFU/mL. Finally, the total volume was adjusted to 200 µL using phosphate-buffered saline. Furthermore, the culture broth was serially diluted using sterile distilled water. From dilutions 4, 5, and 6, a 10 µL suspension was dispensed onto the nutrient agar plate, followed by streaking technique, and incubated at 37 °C overnight. The following day, the antibacterial activities were perceived by counting the CFU from the incubated plates. Positive and negative controls were gentamicin with a known concentration of 100 µg/mL, and PBS, PHA, and PHA/Ch, respectively.

### 2.5. Lactate Dehydrogenase Assays

In brief, HaCaT cells were seeded into the non-pyrogenic 24-well plate with a density of 5 × 10^3^ per well and grown up to 48 h to obtain the confluent monolayer. RPMI-1640 medium and nanocomposites were introduced to monolayer and incubated with 95% humidified, 5% CO_2_, including 37 °C temperature, for 24 h. The release of LDH enzyme into the supernatant was evaluated after the incubation period. However, to evaluate the % cytotoxicity from nanocomposites, we also incubated positive control HAcaT cells treated with 0.1% Triton X-100 and RPMI-1640, including HaCaT cells, as well as a negative control. LDH enzyme release was calculated through % cytotoxicity = sample value − control value/total LDH release − control value × 100. The working principle of LDH assay states that LDH acts as a catalyst for transforming lactate to pyruvate, which produces NADH and H+. In addition, the catalyst diaphorase converts to tetrazolium salt, which reduces to formazan dye. The mechanism between cells and formazan dye color provides quantification of alive and dead cells. From the LDH assay kit, dense purple color signified a high number of viable cells, and light purple color signified cell death.

### 2.6. Characterization

The surface structure and morphology of the PHA/Ch-WS_2_ nanocomposites were analyzed using FE-SEM (Hitachi nanoDUE’T NB5000 USA, Hitachi, Santa Clara, CA, USA). FTIR spectra of the PHA/Ch-WS_2_ nanocomposites were studied using Spectrum RX1 (Perkin Elmer, Waltham, MA, USA) using 4000–400 cm^−1^ range of frequency. Thermogravimetric analysis (TGA) of the samples was obtained by heating the samples from 25 °C to 500 °C at the heating rate of 5 °C/min, under a nitrogen flow (50 mL/min), using a TGA/differential scanning calorimetry 1, Stare System (Mettler Toledo Inc., Columbus, OH, USA).

## 3. Results

### 3.1. Surface Morphology

The fabricated PHA/Ch, PHA/Ch doped with WS_2_ nanocomposite scaffolds and WS_2_ nanoparticles were comprehensively studied by advanced microscopic techniques, FESEM and TEM. The FESEM results of WS_2_ revealed paper-like irregular morphologies on the nanoscale, including the WS_2_ flakes densely stacked onto one another (Figure 2a). Furthermore, the TEM image confirmed the paper-like translucent characteristic of WS_2_ nanoflakes (Figure 2b). The FESEM image of chitosan shown in the previously published work displays a structural morphology similar to cotton flakes, which is the characteristic feature of chitosan [32]. The surface topology obtained from fabricated nanocomposites was evident, along with the development of PHA and Ch polymeric cross-linked network matrix from the early published data [33]. Perhaps the interaction of PHA/Ch macromolecule chains with WS_2_ nanoflakes created the 2D-layered nanocomposite morphology of PHA/Ch-WS_2_ (Figure 2c). Nevertheless, from the FESEM image, it is hard to distinguish the WS_2_ nanoflakes from the matrix, since the nanoflakes were intensely rooted in the polymeric nanocomposite. In addition, tiny concentrations of WS_2_ nanoflakes in the bare polymer matrix were found by EDX, shown in Figure 2d. The EDX investigation confirmed the presence of elements and thus the formation of PHA/Ch-WS_2_.

Comprehensive elemental mapping using SEM image analysis (Figure 3a) appears to be a suitable technique to confirm the elements present in the PHA/Ch-WS_2_ nanocomposite scaffold. Furthermore, their proper distribution, specifically of WS_2_ nanoflakes within the PHA/Ch matrix, is shown in Figure 3b–e. As is apparent from the elemental mapping outcomes, WS_2_ elements (Figure 3d,e) are present homogeneously and consistently in the network of the PHA/Ch matrix (Figure 3b,c) containing carbon and oxygen, indicating the formation of the PHA/Ch-WS_2_ nanocomposite. Figure 3b confirms the presence of elements in the PHA/Ch-WS_2_ nanocomposite.

### 3.2. FT-IR

FT-IR is an effective technique for evaluating the chemical nature of a nanocomposite (Figure 4), in addition to supplementary techniques, such as SEM and XRD. The band numbers of PHA arising at around 1700 cm^−1^ were assigned to the stretching vibration of the C=O group (ester carbonyl), and those at 1150 to 1300 cm^−1^ were assigned to the spectral region associated with the C–O–C bond. A co-polymer of PHA, the HHx band number located at 3436 cm^−1^ was ascribed to the OH vibration (carboxyl group), and the band number at 2933 cm^−1^ was due to –C–H- vibrations of the methylene group [34,35]. The typical bands of chitosan around 3400–3436 cm^−1^ were assigned to the primary amine group and –OH vibration. In chitosan, the bands of amide group C=O were located at 1681 cm^−1^, and the C–H band occurred at 2877 cm^−1^, including C–H bending at 1442 cm^−1^ [36]. The chemical signature of WS_2_, a band at 452, 978 cm^−1^, occurred due to the characteristic S–S bond of sulfur. The band at 604 was ascribed to W-S, and the stretching vibration of the hydroxyl group was assigned at 1442, 1712 cm^−1^ [37].

### 3.3. TGA

Thermal analysis was carried out under the experimental conditions of 20 mL/min nitrogen flow, with a temperature ranging between 30 and 500 °C and an increasing temperature rate of 10 °C/ min. The thermal stability of the synthesized nanocomposites PHA/Ch-WS_2_, PHA/Ch, including bare PHA, Ch, and WS_2_, were evaluated, presented in Figure 5. From the graph, PHA showed minimal loss at around 240 °C due to moisture and impurities, and a drastic mass loss (98.6%) from 268 to 310 °C may be because of free monomers. Chitosan remained until the end of the experiment and exhibited mass loss of 10% and 33% at the ranges between 50–115 °C and 250–343 °C, feasibly due to loss of residual water and the abstraction of side groups, respectively. Furthermore, the PHA/Ch blend exhibited improved thermal stability due to bonding, as compared to their individual forms. The major mass losses of bare PHA and Ch started from 268 to 310 °C and 50 to 343 °C, respectively, whereas the PHA/Ch blend initiated decomposition from 235 to 288 °C. Until the end of the experiment, bare WS_2_ showed a minimal mass loss of 15% and 13% around temperatures of 96–117 °C and 161–232 °C, respectively. This weight loss may be attributed to the loss of moisture, impurities, and the formation of SO_2_ gas, since at high temperatures, O_2_ becomes reactive and may corrode bare WS_2_. PHA/Ch-WS_2_ nanocomposites have demonstrated improved thermal stability due to the incorporation of WS_2_ when compared to the PHA-Ch blend. The nanocomposites’ residue remained (0.2%) until the end of the experiment. The PHA/chitosan and the PHA/Ch-WS_2_ nanocomposites showed similar thermal analysis since the amount of WS_2_ is much less compared to that of the polymer and of chitosan (0.5 wt%). Thus, the obtained data indicate the formation of the PHA/Ch-WS_2_ nanocomposite.

### 3.4. Antibacterial and LDH Assays

To evaluate the antibacterial competence of nanocomposites, different time intervals were assessed, up to 24 h. In the reported time–kill experiment, the most-investigated MDR *E. coli* K1 and MRSA were used to examine their resistance to the nanocomposites. The antibacterial results of three different concentrations of WS_2_ nanocomposites revealed significant (<0.05) bactericidal activity against MDR *E. coli* (Figure 6) and MRSA (Figure 7). From the graph, *E. coli* K1 bactericidal activity of 2 and 4 h compared more strongly to 6 and 24 h. In addition, MRSA was found to be more sensitive at the 2, 4, and 6 h time points compared to the 24 h time point and compared to *E. coli* K1. This difference may be due to the structural changes of Gram-positive and Gram-negative bacteria. Furthermore, the graph demonstrated that the time-dependent and concentration-dependent bactericidal activities could be due to the structure morphology of 3D nanocomposites. However, all three nanocomposites of different WS_2_ concentrations demonstrated a significant reduction in *E. coli* K1 and MRSA for up to 24 h, which suggests that the synthesized nanocomposites are potential scaffolds for antibacterial applications. In addition, the nanocomposites were also tested for their toxicity efficacy against HaCaT cell lines. LDH assay was used to detect cell viability after 24 h of exposure to the nanocomposite. The resulting monolayer was visualized using an inverted microscope at 200×, and the obtained images are presented in Figure 8c. The results demonstrate significant (*p* < 0.05, using an independent *t*-test, two-tailed distribution) cell viability compared to the positive control (Figure 9). However, after 24 h, around 5% damage occurred on the monolayer, which indicated slight toxicity. Increased WS_2_ concentration did not show any major efficacy on the monolayer. In this study, statistical analysis was carried out using two-sample *t*-test distributions, and the *p*-value notations were (*) is *p* < 0.05, (**) is *p* < 0.01, (***) is *p* < 0.001, and error bars designated the standard deviation of triplicate analysis.

## 4. Conclusions

The present research has revealed the successful formation of nanocomposites with a simple solvent casting technique. Furthermore, the results obtained from the characterization, such as FESEM, elemental mapping, and FT-IR, have supported nanocomposites’ development. Cytocompatibility results have demonstrated significant (*p* < 0.05) cell viability up to 24 h with 1.0 wt% wrt PHA, the highest WS_2_ concentration used in this experiment against HaCaT cells, suggesting the nanocomposites are biocompatible. In addition, nanocomposites were revealed to have significant (*p* < 0.05) broad-spectrum antibacterial activities against *E. coli* K1 and MRSA for up to 24 h. Throughout the experiment, nanocomposites were found to be more effective against MRSA than *E. coli* K1. However, the nanocomposites showed significant results. In conclusion, the synthesized nanocomposites were safe and have potential for use in biological applications, specifically as an antibacterial scaffold.

## Figures and Tables

**Figure 1 polymers-14-02224-f001:**
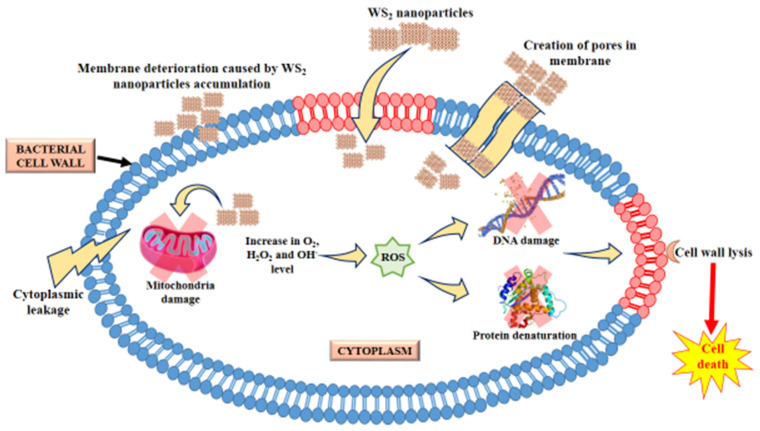
Schematic illustration of nanoparticle-mediated cell death.

**Figure 2 polymers-14-02224-f002:**
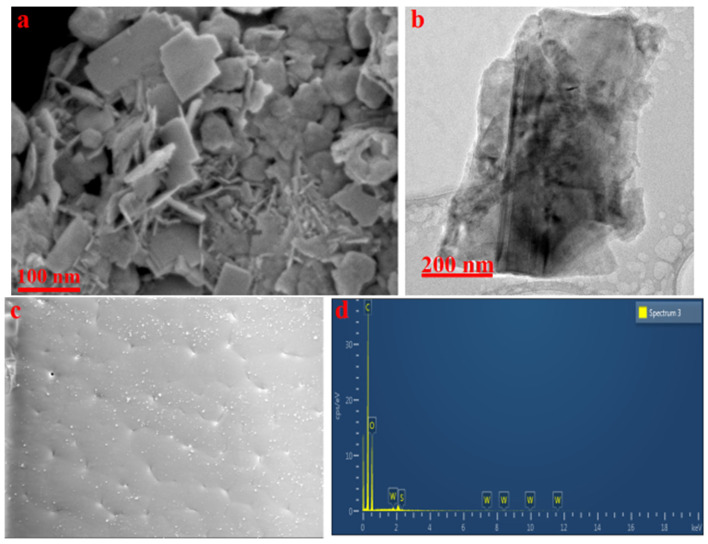
WS_2_ FESEM image. (**a**) TEM image. (**b**) PHA/Ch-WS_2_ nanocomposite. (**c**,**d**) EDX of the PHA/Ch-WS_2_ nanocomposite.

**Figure 3 polymers-14-02224-f003:**
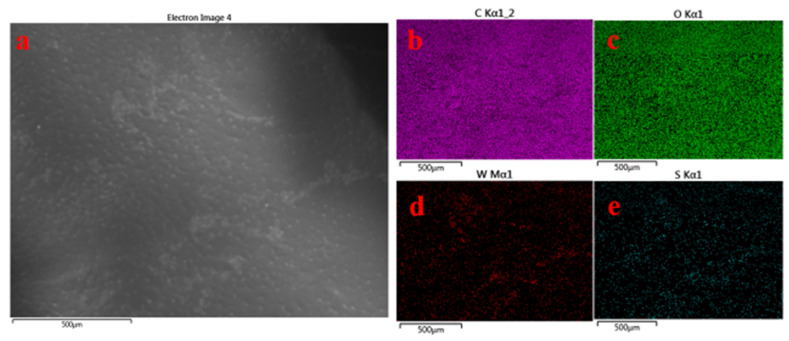
SEM image (**a**–**e**) demonstrates the data of elemental mapping. Image (**a**) is used for mapping, and images (**b**–**e**) represents the elements carbon, oxygen, tungsten, and sulfur, respectively.

**Figure 4 polymers-14-02224-f004:**
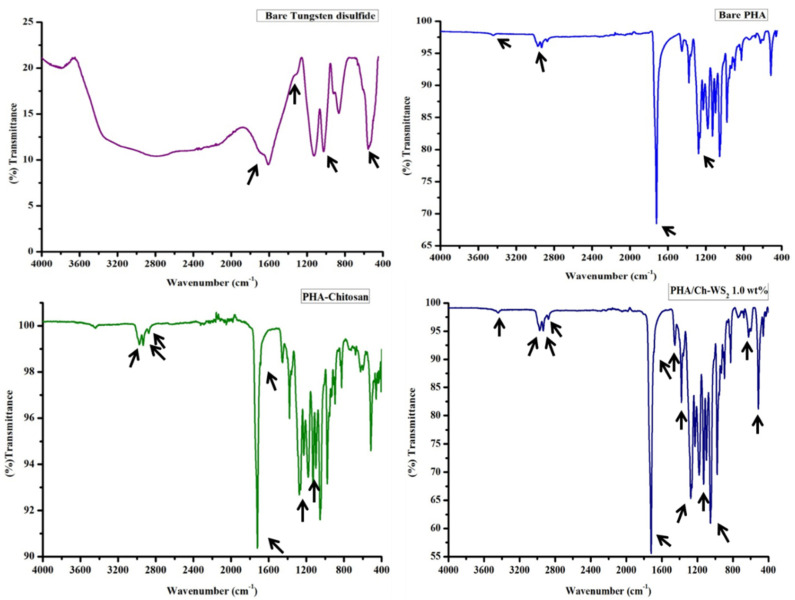
FTIR analysis of bare tungsten disulfide, PHA, PHA-Ch, and PHA-Ch/WS_2_ nanocomposites.

**Figure 5 polymers-14-02224-f005:**
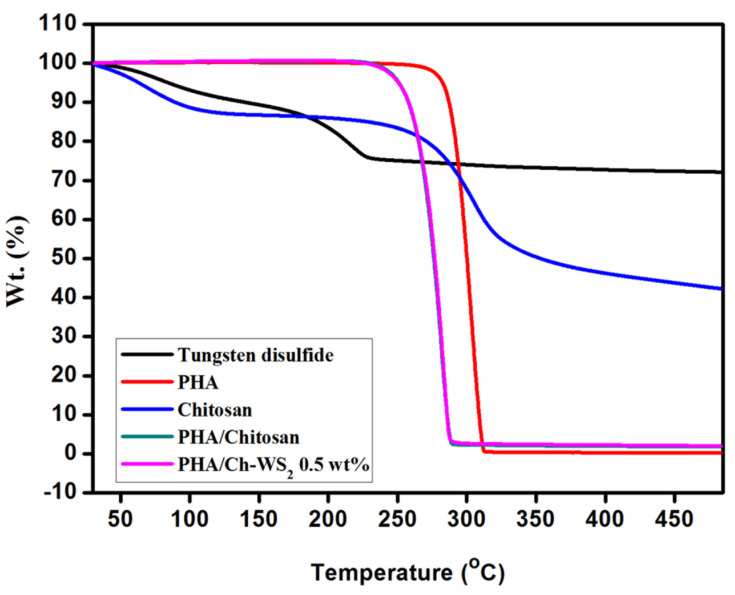
TGA thermogram experimental data of synthesized nanocomposites.

**Figure 6 polymers-14-02224-f006:**
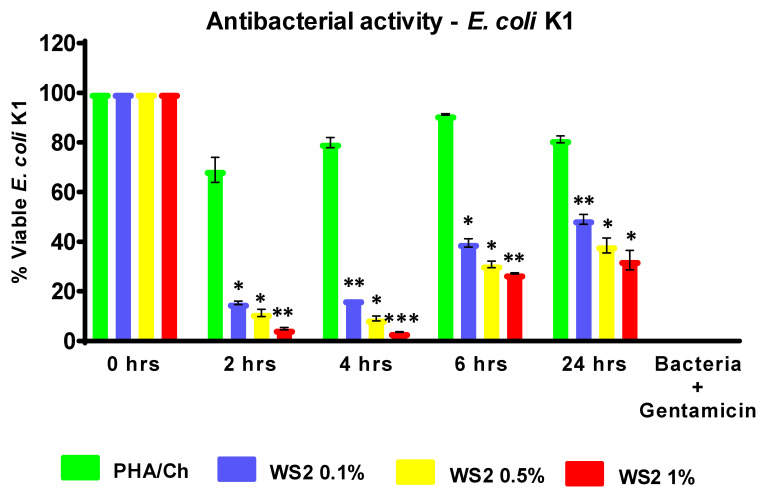
Potential of antibacterial nanocomposites against *E. coli* K1 strain, which exhibited significant antibacterial effects. Statistical analysis obtained by two-sample *t*-test, two-tailed distribution. (*) is *p* < 0.005, (**) is *p* < 0.001, and (***) is *p* < 0.0001.

**Figure 7 polymers-14-02224-f007:**
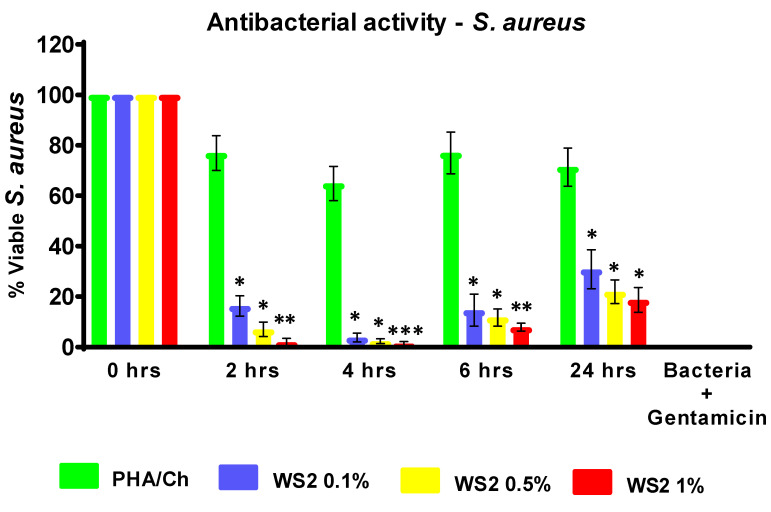
Potential of antibacterial nanocomposites against MRSA strain, which revealed significant antibacterial effects. Statistical analysis attained by two-sample *t*-test, two-tailed distribution. (*) is *p* < 0.005, (**) is *p* < 0.001, and (***) is *p* < 0.0001.

**Figure 8 polymers-14-02224-f008:**
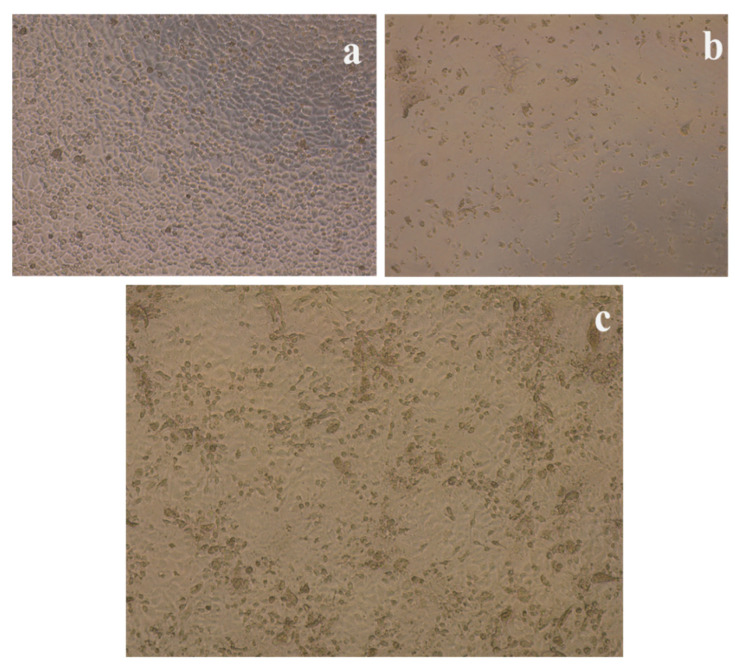
LDH assay to determine HaCaT cell viability against negative (**a**), positive control (**b**), and PHA/Ch-WS_2_ 1% (**c**) nanocomposite, respectively.

**Figure 9 polymers-14-02224-f009:**
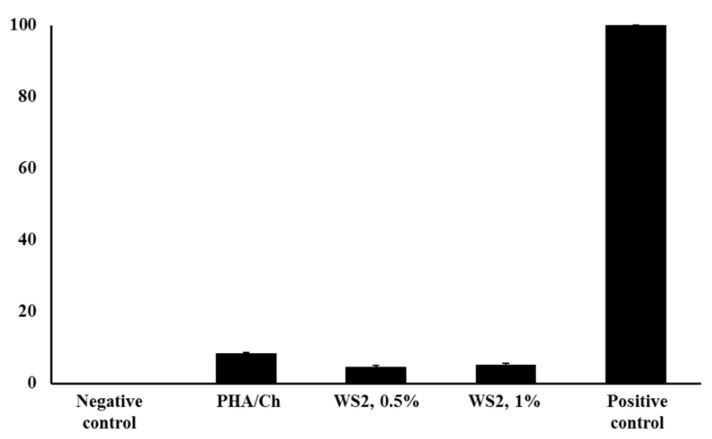
LDH quantitative analysis of cell viability against nanocomposites.

## Data Availability

Not Applicable.

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
