# Peer review of "Development of Biocompatible Polyhydroxyalkanoate/Chitosan-Tungsten Disulphide Nanocomposite for Antibacterial and Biological Applications"

_polymers, 2022, doi:10.3390/polym14112224_

Round 1

Reviewer 1 Report

In this article the authors prepared the biocompatible polyhydroxyalkanoate/chitosan-tungsten disulphide nanocarriers and investigated them for antibacterial and other biological applications

These are the following major comments as follows

  1. The overview presentation of Fig.1. is not related to nanocarriers delivery. It is recommended to improve this fig.1. and show the possible nanoparticle penetration inside the bacterial cell wall.
  2. In the introduction section, L-96, it is recommended to include these references which are related to chitosan and polysaccharides. A. Appraisal of Chitosan-Gum Arabic-Coated Bipolymeric Nanocarriers for Efficient Dye Removal and Eradication of the Plant Pathogen Botrytis cinerea. ACS Applied Materials & Interfaces. 2021 Oct 1;13(40):47354-70. Recent findings and future directions of grafted gum karaya polysaccharides and their various applications: A review. Carbohyd. Polym.. 2021 Apr 15;258:117687. C. Chitosan-gum arabic embedded alizarin nanocarriers inhibit biofilm formation of multispecies microorganisms. Carbohyd. Polym.. 2022 May 15;284:118959.
  3. In the last section of introduction, it is recommended to be included the main rationale of research.
  4. In the preparation section 2.2. authors have not mentioned the % of glacial acetic acid which was used for the dissolving chitosan.
  5. In the SEM image, it is not observed particle of nanocomposite. If author have high resolution image of SEM then replace image Fig.3a.
  6. The Y axis must be increased in both Fig 6 and 7.

Author Response

Reviewer#1: In this article the authors prepared the biocompatible polyhydroxyalkanoate/chitosan-tungsten disulphide nanocarriers and investigated them for antibacterial and other biological applications. These are the following major comments as follows:

Response: We thank the Reviewer #1, for the valuable comments on our manuscript. These comments helped us to improve the manuscript quality and presentation style. Please refer below to responses for the reviewer comments.

Comment 1: The overview presentation of Fig.1. is not related to nanocarriers delivery. It is recommended to improve this fig.1. and show the possible nanoparticle penetration inside the bacterial cell wall.

Response: The figure is just a simple illustration of the proposed cell death  

Comment 2: In the introduction section, L-96, it is recommended to include these references which are related to chitosan and polysaccharides. A. Appraisal of Chitosan-Gum Arabic-Coated Bipolymeric Nanocarriers for Efficient Dye Removal and Eradication of the Plant Pathogen Botrytis cinerea. ACS Applied Materials & Interfaces. 2021 Oct 1;13(40):47354-70. Recent findings and future directions of grafted gum karaya polysaccharides and their various applications: A review. Carbohyd. Polym.. 2021 Apr 15;258:117687. C. Chitosan-gum arabic embedded alizarin nanocarriers inhibit biofilm formation of multispecies microorganisms. Carbohyd. Polym.. 2022 May 15;284:118959.

Response: We thanks the reviewer for his suggestion. The suitable references have been added as per reviewer’s suggestion.

Comment 3: In the last section of introduction, it is recommended to be included the main rationale of research.

Response: As suggested by the reviewer the main rationale of this study have been added in last paragraph of the introduction section.

Comment 4: In the preparation section 2.2. authors have not mentioned the % of glacial acetic acid which was used for the dissolving chitosan.

Response: We thanks the reviewer for his comment and indicating this error. As suggested, the % of glacial acetic acid with weight ratio of chitosan has been added in the revised version of the manuscript.

Comment 5: In the SEM image, it is not observed particle of nanocomposite. If author have high resolution image of SEM then replace image Fig.3a.

Response: This is an image for the elemental mapping which is not discussed to morphological analysis. This image is only representing a selected area for elemental mapping. At this point, we do not have any image with higher resolution. We will keep reviewers suggestion in mind and take high resolution images during elemental mapping also.

Comment 6: The Y axis must be increased in both Fig 6 and 7.

Response: As suggested by the reviewer, the Y axis in both fig. 6 ans 7 has been increased and the figures have been redrawn again.

Reviewer 2 Report

  1. In Page2, line 58.  What is MDR model? Can the author provide the full name in English?
  2. During the film casting process, will PHA dissolve in high temperature glacial acetic acid, will it affect its molecular weight?
  3. In Figure 5, the sample PHA was analyzed using raw material or cast film? On the other hands, the TGA curve of sample PHA/Chitosan and PHA/Ch-WS2 0.5% is completely overlapping. I think the data may be misplaced, please check again by the author.
  4. In Page 7, Line 244. The under sentence, “Thermal analysis was carried out under the experimental conditions of 20 mL/min nitrogen flow, temperature ranging between 30-500oC with an increasing temperature rate of 10oC/min.”, where the unit of temperature should be in superscript, not lowercase o.
  5. In Section 2.2, The  WS2 weight ratio in the PHA/CS-WS2 0.5wt% composite was calculated is about 0.05wt% (when 0.5mg of WS2 nanoparticles was added) .  However, the thermal analysis results of the sample PHA/CS-WS2 0.5wt% showed that the residue (ash) was 0.2%. Since the ash ratio is related to the content of WS2 in the sample. The analysis results do not match the experimental formula.  The preparation process of the sample and the labelling need to be reconfirmed.
  6. In the Fig 6 and Fig 7, It can be seen that the antibacterial results are positively correlated with the amount of WS2 added. Is the result related to the ratio of exposed WS2 nanoparticles on the composite surface?

Author Response

Reviewer #2:

Comment 1: In Page2, line 58.  What is MDR model? Can the author provide the full name in English?

Response: The full name of MDR is provided in the revised manuscript. See page 2 line 59.

Comment 2: During the film casting process, will PHA dissolve in high temperature glacial acetic acid, will it affect its molecular weight?

Response: The films are dissolved at a temperature which has been reported to be an optimized process.[1] In general, at low casting temperatures, the solvent evaporates slowly, and there is limited thermal energy available for crystallization. Films, therefore, have low crystallinity, have good mechanical properties (in terms of tensile strength and strain to failure), and reasonable optical transmittance. However, films processed at lower temperatures have rougher surfaces both at the macroscopic and microscopic scale – due to variations in thickness resulting from phase separation, and to inhomogeneities caused by the limited thermal energy available for crystallization. On the other hand, higher solvent casting temperatures yield films that are more crystalline, more transparent, and have higher surface uniformity. However, these films have relatively low tensile strength and strain. These results show that the proper selection of casting temperature and solvent evaporation rate can be used to achieve films with the desired set of properties. Thus, 118 is an optimum temperature where the obtained film is having good physical and chemical properties.

[1]. Anbukarasu, P., Sauvageau, D., & Elias, A. (2015). Tuning the properties of polyhydroxybutyrate films using acetic acid via solvent casting. Scientific reports, 5(1), 1-14.

Comment 3: In Figure 5, the sample PHA was analyzed using raw material or cast film? On the other hands, the TGA curve of sample PHA/Chitosan and PHA/Ch-WS2 0.5% is completely overlapping. I think the data may be misplaced, please check again by the author.

Response: We thanks the reviewer for his comment on TGA. Since the amount of WS2 is very less as compared to PHA/chitosan (mere 0.5%), hence the change is the TGA curve is not visible apparently. There is only a slight increase in thermal properties which is reflected in the TGA curve (0.2wt% residue has remained (0.2%) until the end of the experiment). However, in lieu to the reviewer’s comment we have mentioned this fact in the revised version of the manuscript.

Comment 4: In Page 7, Line 244. The under sentence, “Thermal analysis was carried out under the experimental conditions of 20 mL/min nitrogen flow, temperature ranging between 30-500oC with an increasing temperature rate of 10oC/min.”, where the unit of temperature should be in superscript, not lowercase o.

Response: We thanks the reviewer for pointing this typo error. Ass suggested the unit has been corrected in revised version of the manuscript.

Comment 5: In Section 2.2, The WS2 weight ratio in the PHA/CS-WS2 0.5wt% composite was calculated is about 0.05wt% (when 0.5mg of WS2 nanoparticles was added) .  However, the thermal analysis results of the sample PHA/CS-WS2 0.5wt% showed that the residue (ash) was 0.2%. Since the ash ratio is related to the content of WS2 in the sample. The analysis results do not match the experimental formula.  The preparation process of the sample and the labelling need to be reconfirmed.

Response: As can be seen from the TGA thermogram of the WS2, there is some weight loss (28 wt% weight loss), which may be due to formation of SO2 at high temperature. Since at high temperature O2 become highly reactive and can corrode WS2 by formation of SO2 etc. Thus, this weight loss is also reflected in the TGA of the nanocomposites. Also, during loading of WS2, some of the unbound WS2, may leach out while washing. However, as pointed by reviewer, we have included these explanation in TGA section of the revised version of the manuscript.

Comment 6: In the Fig 6 and Fig 7, It can be seen that the antibacterial results are positively correlated with the amount of WS2 added. Is the result related to the ratio of exposed WS2 nanoparticles on the composite surface?

Response: We thanks the reviewer for his comment. Results from antibacterial assays revealed that WS2 presented concentration dependent antibacterial activity against both the Gram-negative (E. coli) and Gram-positive (MRSA) as shown in Figure 6 and 7 respectively. It suggests that the bioactivity might be directly linked with the WS2 embedded in the nanocomposite. The ratio of the exposed WS2 in the nanocomposites is the aim for the future studies.

Reviewer 3 Report

The manuscript titled “Development of Biocompatible Polyhydroxyalkanoate/Chitosan-Tungsten Disulphide Nanocomposite Towards the Antibacterial and Biological Applications” by Mukheem et al is a research paper mainly investigated the antibacterial property and biocompatibility of the 2D PHA/Ch-WS2 nanocomposite. The designed material showed no obvious cytotoxicity to cells but exhibited impressive antimicrobial ability against both gram-negative and -positive species. This finding is meaningful. However, the data has not been well presented and the underlying mechanisms have not been well revealed. Besides, the poor written further significantly reduces my enthusiasm.

  1. Countless language problems, please check the manuscript thoroughly.

e.g.      Line 21, improvise—improve? reduces—reduce.

Line 25, was—is

Line 26, organism—organisms

Line 40, contains—contain

Lines 46—47

Lines 52—53

Also, please try to use concise languages. For example, Lines 93—97, the introduction of Chitosan is tedious.

  1. Line 20, what common features they share should be briefly specified.

  1. The resolution for Fig. 2 is too low.

Please remove the bottom edges of 2a/c and make scale bars manually. You don’t need to provide other information. It shows these pictures were taken a few years ago and it may reduce the timeliness and novelty of your work. Please change the color or position of your scale bar for 2b as it stacks together with the edge of your NPs. In 2d, the peaks for W are barely seen. Is it possible to provide a quantitative data here from EDS?

  1. Fig. 3b is whole black? Why don’t W and S signals show any overlap?

  1. What is the thickness of your 2D material? If the prepared film is too thin, the elemental mapping signals could also come from the substrate. Line 151 only shows the size of the substrate used.

  1. The characterization section is missing in “Materials and Methods”.

  1. Line 31, why is “environment” mentioned here? This will cause confusion to readers.

  1. Is there any evidence demonstrating your explanation regarding the antibacterial mechanism of your material exhibited in Fig. 1? For example, have you tested the ROS levels? Or is it possible to provide any FESEM images showing that the membranes of E. coli and S.A. are destroyed by the sharp morphology of your material?

Author Response

Reviewer #3: The manuscript titled “Development of Biocompatible Polyhydroxyalkanoate/Chitosan-Tungsten Disulphide Nanocomposite Towards the Antibacterial and Biological Applications” by Mukheem et al is a research paper mainly investigated the antibacterial property and biocompatibility of the 2D PHA/Ch-WS2 nanocomposite. The designed material showed no obvious cytotoxicity to cells but exhibited impressive antimicrobial ability against both gram-negative and -positive species. This finding is meaningful. However, the data has not been well presented and the underlying mechanisms have not been well revealed. Besides, the poor written further significantly reduces my enthusiasm.

Comment 1: Countless language problems, please check the manuscript thoroughly.

e.g.      Line 21, improvise—improve? reduces—reduce.

Line 25, was—is

Line 26, organism—organisms

Line 40, contains—contain

Lines 46—47

Lines 52—53…

Also, please try to use concise languages. For example, Lines 93—97, the introduction of Chitosan is tedious.

Response: We thanks the reviewer for pointing out the typo errors. All the suggestions have been incorporated and entire manuscript have been proof read to remove any typo error or grammatical mistakes.

Comment 2: Line 20, what common features they share should be briefly specified.

Response: As suggested by the reviewer, we have added some common features in the revised version of the manuscript.

Comment 3: The resolution for Fig. 2 is too low.

Please remove the bottom edges of 2a/c and make scale bars manually. You don’t need to provide other information. It shows these pictures were taken a few years ago and it may reduce the timeliness and novelty of your work. Please change the color or position of your scale bar for 2b as it stacks together with the edge of your NPs. In 2d, the peaks for W are barely seen. Is it possible to provide a quantitative data here from EDS?

Response: We thanks the reviewer for his constructive comments. These comments have helped us a lot to improve the quality of the manuscript. As suggested by the reviewer, the following changes has been done. However, we have repeated EDS many times, but got W peaks which are having low intensities. The main reason is the amount of is just 2 wt.% w.r.t. to PHA which is very low, hence peaks are of low intensities.

Comment 4: Fig. 3b is whole black? Why don’t W and S signals show any overlap?

Response: we have tried to perform the elemental analysis, but every time, the compiled image of all elements obtained in PHA/Ch-WS2 nanocomposite appears to be black. This may be either due to the burning of the sample or overlapping producing very poor contrast. In lie to the reviewer’s comments, we have removed figure 3b so that it may not create any confusion.

Comment 5: What is the thickness of your 2D material? If the prepared film is too thin, the elemental mapping signals could also come from the substrate. Line 151 only shows the size of the substrate used.

Response: The film is 1-5 mm in thickness. It was not too thin so as the elemental mapping signals could also come from the substrate.

Comment 6: The characterization section is missing in “Materials and Methods”.

Response: As suggested by the reviewer, the characterization section has been incorporated in the revised version of the manuscript.

Comment 7: Line 31, why is “environment” mentioned here? This will cause confusion to readers.

Response: We thanks the reviewer for his keen evaluation of our manuscript. We agree with the reviewer’s opinion that environment is a confusing term. We have replaced this by ‘biodegradable’, since this was the property we want to mention in this part.

Comment 8: Is there any evidence demonstrating your explanation regarding the antibacterial mechanism of your material exhibited in Fig. 1? For example, have you tested the ROS levels? Or is it possible to provide any FESEM images showing that the membranes of E. coli and S.A. are destroyed by the sharp morphology of your material?

Response: In the present study, we have evaluated the WS2 nanocomposites for the in vitro antibacterial activity against E. coli and MRSA. The results have shown that WS2 exhibited effective antibacterial effects. In future, our group is working on the exact mechanism of action as well as the In vivo activity using laboratory mice model. Previously, in a study WS2 nanosheets showed strong antibacterial activity against E. coli and Staphylococcus aureus by damaging the bacterial membrane (Liu et al., 2017). Based on the literature we can speculate that the WS2 tested in this study might have the same effects against these MDR bacteria isolates.

[1] Liu, X., Duan, G., Li, W., Zhou, Z. and Zhou, R., 2017. Membrane destruction-mediated antibacterial activity of tungsten disulfide (ws 2). Rsc Advances, 7(60), pp.37873-37880.

Round 2

Reviewer 1 Report

The revised manuscript is suitable for further process in this journal. 

Reviewer 2 Report

no comments

Reviewer 3 Report

The manuscript has been significantly improved and my concerns have been well addressed.